# Age of first diagnosis and incidence rate of uterine fibroids in Ghana. A retrospective cohort study

Emmanuel Kobina Mesi Edzie[1]*, Klenam Dzefi-Tettey[2], Edmund Kwakye Brakohiapa[3], Frank Quarshie[4], Sebastian Ken-Amoah[5], Obed Cudjoe[6], Evans Boadi[7], Joshua Mensah Kpobi[7], Richard Ato Edzie[1], Henry Kusodzi[1], Prosper Dziwornu[1], Abdul Raman Asemah[1]

1 Department of Medical Imaging, School of Medical Sciences, College of Health and Allied Sciences, University of Cape Coast, Cape Coast, Ghana, 2 Department of Radiology, Korle Bu Teaching Hospital, Accra, Ghana, 3 Department of Radiology, University of Ghana Medical School, Accra, Ghana, 4 African Institute for Mathematical Sciences (AIMS), Summerhill Estates, East Legon Hills, Santoe, Accra, Ghana, 5 Department of Obstetrics and Gynecology, School of Medical Sciences, College of Health and Allied Sciences, University of Cape Coast, Cape Coast, Ghana, 6 Department of Microbiology and Immunology, School of Medical Sciences, College of Health and Allied Sciences, University of Cape Coast, Cape Coast, Ghana, 7 Department of Radiology, Cape Coast Teaching Hospital, Cape Coast, Ghana

* emmanuel.edzie@ucc.edu.gh

**Data Availability Statement:** All relevant data are within the paper and its Supporting Information files.

## Abstract

### Background

Uterine fibroids are benign tumors that grow in or on the uterus of women. Globally, they occur in more than 80% of women of African ancestry and 70% in white women. Uterine fibroid requires much attention because of its high incidence rate among women of all races and ages. This study sought to document the age of first diagnosis and incidence rates of uterine fibroids in our urban setting.

### Methods

This study reviewed and analyzed the ages and year of diagnosis of all 2,469 patients with the first diagnosis of uterine fibroids from 1st January 2018 to 31st December 2021 in South-Central Ghana. The obtained data were analyzed using GNU PSPP, Python on Jupyter Notebook and Libre Office Calc with statistical significance level set at $p \leq 0.05$.

### Results

The overall average age was 36.29±8.08 years, with age range 17–61 years and the age groups with the highest frequencies were 35–39 years (n = 642, 26.00%), 30–34 years (n = 563, 22.80%) and 40–44 years (n = 381, 15.43%). The mean ages of the patients in 2018, 2019, 2020 and 2021 were 36.70±8.00 years (95%CI = 35.97–37.43), 37.07±7.66 years (95%CI = 36.45–37.70), 35.92±7.87 years (95%CI = 35.30–36.53) and 35.78±8.54 years (95%CI = 35.19–36.38) respectively. The incidence rate (per 100,000) of uterine fibroids in 2018, 2019, 2020 and 2021 were 66.77 (95% CI = 60.63–72.90), 81.86 (95%CI = 75.19–

**Funding:** The author(s) received no specific funding for this work.

**Competing interests:** The authors have declared that no competing interests exist.

88.58), 85.60 (95%CI = 78.85–92.35) and 92.40 (95%CI = 85.88–98.92) respectively, with 35–39 age group recording the highest in all years.

## Conclusion

The incidence rate of uterine fibroids increased as the years progressed and it is mostly high in 35–39 years age category, with a decreasing annual mean age trend indicative of early diagnosis.

## Introduction

Uterine fibroids (leiomyomas/fibroids) are benign tumors that grow in or on the uterus of women, mostly during their childbearing age. They are made of fibrous connective tissues and smooth muscle cells, which are associated with pelvic pain or pressure, anemia, infertility, abnormal uterine bleeding, and back pain, but usually asymptomatic in about 30% of women as reported in some studies [1–3]. Globally, uterine fibroids are seen in more than 80% of women of African ancestry and 70% of white women [3, 4].

The diagnoses of uterine fibroids are usually made using clinical presentations, medical imaging modalities like ultrasonography, magnetic resonance imaging (MRI), and then histopathology for confirmation [5–8]. A recent study in Ghana has reported ultrasound as the most available imaging modality in radiological practices. This study also reported the availability of MRI in the country [9]. However, there is an increased use of ultrasound equipment because it is the first-line imaging modality for the diagnosis of uterine fibroids and pregnancy, as it is inexpensive, easier to use, very portable and clearly provides adequate features for the diagnosis of uterine fibroids [10–12].

The etiology of uterine fibroids is unknown but, is associated with the following risk factors; hormonal factors, obesity, race, age, and some lifestyle activities like smoking, diet, stress, alcohol, and caffeine consumption [13]. Treatment options for uterine fibroids are medical, surgical (myomectomy, hysterectomy), and interventional (uterine artery embolization, and magnetic resonance-guided focused ultrasound surgery) [7, 14].

A study by Stewart et al. showed that the incidence of uterine fibroids is high ranging from 217 to 3,745 per 100,000 women in America, and prevalence, also spanning from 4.5 to 68.6% across the continent (Europe, Asia, Africa, North and South America) [4]. The variations in the incidence and prevalence rates were attributed to the differences in the method of diagnosis and study population [15]. Literature posits a high incidence and prevalence of uterine fibroids in black women as compared to white women [15–18]. Wechter et al. reported that the incidence rates for black women are 2- to 3-fold higher than for white women [16]. The prevalence rate for black women was also high (18.5%) as compared to that of white women (10.3%) [17]. A study by Igboeli et al., to highlight the burden of uterine fibroids in Sub-Saharan Africa, revealed that, 70–80% of black women will harbor uterine fibroids over their lifetime, thereby calling for action and opportunity for intervention [19].

In Ghana, a report shows that uterine fibroids are common, however, there are limited studies on the incidence of this condition. Uterine fibroids have significant social, economic and, medical implications for the female populace in Ghana. Literature has revealed that, despite the reproductive desires of Ghanaian women with uterine fibroids, about 40% of them are likely to have a hysterectomy [20, 21]. Findings from Ofori-Dankwa et al., have also highlighted an increase in medical costs, resulting in a high disease burden of uterine fibroids

in Ghana [22]. Information on the age of first diagnosis and incidence rate of uterine fibroids are extremely crucial in laying out interventions to manage this disorder. Nonetheless, such information is scanty, hence this study, to determine the age of first diagnosis and the incidence rate of uterine fibroids. The following were the specific objectives considered;

- To determine the mean age of the first diagnosis of uterine fibroids and any possible changing trends over the study period.

- To ascertain whether there is a difference in the mean age of first-diagnosed uterine fibroids over the study period.

- To assess the annual incidence rates of uterine fibroids and the trends.

## Materials and methods

### Study site and design

The central region of Ghana is one of the sixteen regions in the country with a population of about 2.86 million representing 9.29%, according to the Ghana Statistical Service report on the 2021 population and housing census. The report showed that the females are 2.7% more than the males, with the population of the males being 1,390,985 and that of the females as 1,468,836. Similar differences in population sizes across the sexes in the region were recorded in 2018, 2019 and, 2020 [23, 24]. This study was conducted in the Cape Coast Teaching Hospital (CCTH), which is currently the only public tertiary health facility situated in Cape Coast, the regional capital serving all the districts in the entire region and beyond, hence the most endowed facility in terms of human and material resources for delivering tertiary services including advanced pathology and radiology services. We carried out a population-based retrospective cohort study on females who had been enrolled in the Lightwave Health Information Management System (LHIMS) with complete data for the most recent 4-year period (Jan. 1, 2018, through Dec. 31, 2021). In reference to the available cases from the records, the cohort was made up of all females aged 17 to 61 years in 2018 through 2021 who had been enrolled in the LHIMS for at least three years, and have no history of uterine fibroids, myomectomy, and hysterectomy three years prior to the cohort year. This retrospective cohort study categorized the ages of the patients as "15–19 years", "20–24 years", "25–29 years", "30–34 years", "35–39 years", "40–44 years", "45–49 years", "50–54 years", "55–59 years", and "60–64 years", based on the World Health Organization (WHO) age categorization, to ascertain the incidence rates for the various age groups [25].

### Data collection

We retrospectively retrieved and reviewed the records of all patients diagnosed for the first time with uterine fibroids via ultrasonography and histo-pathology from 1st January, 2018 to 31st December, 2021, as the diagnosis of uterine fibroids is mostly done by ultrasonography, and histology. We used the Lightwave Health Information Management System (LHIMS), which is available in all health facilities in the region including the CCTH, and therefore information obtained from this electronic system is reflective of the study population. The LHIMS contains all and detailed electronic health records of patients. The richness and endurance of the many computerized databases in CCTH are standout characteristics, which have been extensively used for many years for research and patient care. Detailed information on demographics, enrollment, weight, height, procedures, diagnosis, pharmacy dispensing, laboratory and radiology results have been kept in automated databases. The LHIMS also contains patients' contact details, clinic notes, and past medical history. Keywords such as uterine

myomas, fibroids, leiomyomas, hysterectomy, and myomectomy were used to obtain all the patients who have been diagnosed with uterine fibroids within the cohort years. Their unique electronic identification numbers were obtained afterwards. These were done by three of the researchers who are practicing doctors in CCTH (each with a unique code for accessing the LHIMS). Using the unique electronic identification number obtained, all automated data sources of the patients which are linked can be retrieved. Women who had no fibroid diagnosis in the three years prior to the study cohort year were eligible for inclusion. Patients with past medical history of uterine fibroids, hysterectomy, and myomectomy from the electronic records and those whose previous gynecological records for at least 3 years prior to the study cohort year could not be obtained were excluded from the study. All patients (who met the above inclusion criteria) with the first diagnosis of uterine fibroids over the period were purposively selected and included. In order to ensure the inclusion of only patients with first diagnosis of uterine fibroids, and avoid duplication of participants, researchers did a detailed review of the medical records with the help of patients' unique electronic identification numbers. A total of 2469 patients who met the inclusion criteria for first-time diagnosed uterine fibroids were retrieved for extensive analyses.

## Statistical analysis

We found the annual specific incidence rates of first-diagnosed uterine fibroids as a ratio of the number of events to the projected total population in the region respectively for 2018, 2019, 2020, and 2021. The population projections were done by the Ghana Statistical Service. To overcome the variation effects in age structure, adjustments of age-specific rates were done using the direct method of standardization founded on the WHO population distribution standard. Poisson distribution was used to estimate the incidence rates with a 95% confidence interval (CI). The significant difference in the annual mean age of first-diagnosed uterine fibroid patients was tested using the one-way Analysis of Variance (ANOVA), the Robust test, and the F-test for equality of means. The Games-Howell's test was also used for multiple comparisons. Preliminary analyses were carried out to check for violation of assumptions particularly (the Poisson distribution test, homogeneity of variance, and normality) for analyses, where applicable. All two-sided p-values $\leq$ 0.05 were considered statistically significant. Mean values were quoted as absolute values ±standard deviation. The analyses were done using the GNU PSPP (Category: Education, Science & Math), pspp version 1.2.0–3, developed by the Free Software Foundation, Python (developed by Python Software Foundation, version 3) on Jupyter Notebook, and the LibreOffice Calc, developed by The Document Foundation (version 1:6.1.5–3+deb10u6).

## Ethical consideration

The Cape Coast Teaching Hospital Ethical Review Committee (CCTHERC) gave approval for this research to be carried out, with clearance number, CCTHERC/EC/2020/102. This was a retrospective study over a four-year period. Informed consent was not obtained for this large number of patients as we could not have contacted all the patients because patients might have changed contacts, dead at the time of data collection, or been long discharged. These were made known to the CCTHERC and no concerns were raised. Confidentiality and anonymity were ensured throughout the study. This study conformed to the 1975 Declaration of Helsinki.

## Results

The overall average age of the 2,469 patients with the first diagnosis of uterine fibroids was 36.29±8.08 years with an age range of 17–61 years. The age group with the highest frequency

**Table 1. Socio-demographics of the participants and annual mean age comparison.**

| Variable | | | | | | |
|---|---|---|---|---|---|---|
| | | Minimum | Maximum | Mean (SD) | | |
| Age (Overall) | | 17 | 61 | 36.29 (8.08) years | | |
| **Age Group** | **2018** | **2019** | **2020** | **2021** | **Total** | **P-value** |
| 15–19 | 4(0.86%) | 6(1.04%) | 8(1.27%) | 13(1.64%) | 31(1.26%) | 0.004* |
| 20–24 | 24(5.16%) | 16(2.76%) | 29(4.60%) | 52(6.55%) | 121(4.90%) | |
| 25–29 | 56(12.04%) | 68(11.74%) | 98(15.53%) | 122(15.37%) | 344(13.93%) | |
| 30–34 | 112(24.09%) | 136(23.49%) | 138(21.87%) | 177(22.29%) | 563(22.80%) | |
| 35–39 | 109(23.44%) | 144(24.87%) | 169(26.78%) | 220(27.71%) | 642(26.00%) | |
| 40–44 | 83(17.85%) | 107(18.48%) | 102(16.16%) | 89(11.21%) | 381(15.43%) | |
| 45–49 | 46(9.89%) | 55(9.50%) | 47(7.45%) | 55(6.93%) | 203(8.22%) | |
| 50–54 | 21(4.52%) | 42(7.25%) | 29(4.60%) | 47(5.92%) | 139(5.63%) | |
| 55–59 | 9(1.94%) | 4(0.69%) | 9(1.43%) | 17(2.14%) | 39(1.56%) | |
| 60–64 | 1(0.22%) | 1(0.17%) | 2(0.32%) | 2(0.25%) | 6(0.24%) | |
| Total | 465(100%) | 579(100%) | 631(100%) | 794(100%) | 2469(100.00%) | |
| **Annual Mean Age Comparison** | | | | | | |
| Year | **Mean** | **SD** | **95% CI for Mean** | | **F-value** | **_P_-value** |
| 2018 | 36.7 | 8 | 35.97–37.43 | | 3.726 | 0.011* |
| 2019 | 37.07 | 7.66 | 36.45–37.70 | | | |
| 2020 | 35.92 | 7.87 | 35.30–36.53 | | | |
| 2021 | 35.78 | 8.54 | 35.19–36.38 | | | |
| Overall | 36.29 | 8.08 | 35.97–36.61 | | | |

CI = Confidence Interval; SD = Standard Deviation

*Statistically Significant

was 35–39 years ($n$ = 642, 26.00%), followed by 30–34 years ($n$ = 563, 22.80%) and 40–44 years ($n$ = 381, 15.43%). The least populated age category was 60–64 years ($n$ = 6, 0.24%). The remaining results are shown in (Table 1). The mean ages of the patients with uterine fibroids in 2018, 2019, 2020, and 2021 were 36.70±8.00 years (95% CI = 35.97–37.43), 37.07±7.66 years (95% CI = 36.45–37.70), 35.92±7.87 years (95% CI = 35.30–36.53) and 35.78±8.54 years (95% CI = 35.19–36.38) respectively. Statistically, there were significant differences in the annual average ages of the patients ($F$-value = 3.726, $p$-value = 0.011) as also shown in (Table 1).

The annual mean ages of the included patients with uterine fibroids showed an initial surge in 2019 and declined afterwards across the years as depicted in (Fig 1).

Post hoc tests carried out revealed that the mean age of the patients in 2018 was statistically the same as the other years (2019, 2020 and 2021). However, there were significant differences between the average age of the patients in 2019 and that of 2020 and, 2021, with mean differences of 1.157 (95% CI = 0.01–2.31, $p$-value = 0.048), and 1.292 (95% CI = 0.16–2.42, p-value = 0.018) respectively. The rests are displayed in (Table 2).

The incidence rate (IR) of uterine fibroids in 2018 was 66.77 per 100,000 (95% CI = 60.63–72.90) with 35–39 years age category recording the highest (IR = 15.58 per 100,000; 95% CI = 12.66–18.50), followed by the 30–34 years (IR = 15.35 per 100,000; 95% CI = 12.51–18.20) and the 40–44 years (IR = 12.29 per 100,000; 95% CI = 10.18–15.76) age groups, with 60–64 years age group recording the lowest (IR = 0.19 per 100,000; 95% CI = -0.18–0.56) (Table 3). The incidence rate in 2019 was 81.86 per 100,000 (95% CI = 75.19–88.58) with 35–39 years age class having the highest rate (IR = 20.16 per 100,000; 95% CI = 16.87–23.45), and 60–64 years

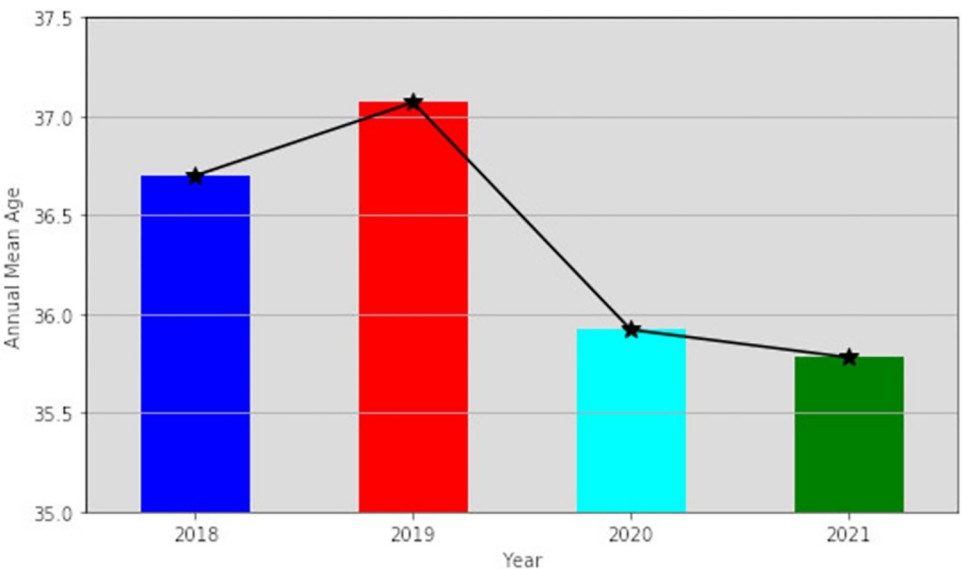

**Fig 1. Annual mean age of the patients with first diagnosis of uterine fibroids.**

age category recording the least (IR = 0.19 per 100,000; 95% CI = -0.18–0.55) as shown in (**Table 3**).

In 2020, the overall incidence rate was 85.60 per 100,000 (95% CI = 78.85–92.35) and that of 2021 was 92.40 per 100,000 (95% CI = 85.88–98.92). The contribution of the age categories to the overall annual incidence rates in 2020 and 2021 are shown in (**Table 4**).

Fig 2 shows the graphical representation of the age-adjusted incidence rates of uterine fibroids over the study period. It depicts a high incidence of uterine fibroids in the 35–39 years age group across all the cohort years examined (2018, 2019, 2020, and 2021). The incidence rate of uterine fibroids in 2021 was the highest across the age groups (as compared to the other years) with the exception of 40–44 and 45–49 age categories, where the incidence in 2019 was the highest (**Fig 2**).

**Table 2. Post hoc tests on the significant results from the analysis of variance (ANOVA).**

| Year (I) | Year (J) | Mean Diff | 95% CI for Mean Diff | P-value |
|---|---|---|---|---|
| 2018 | 2019 | -0.371 | -1.63–0.89 | 0.873 |
|  | 2020 | 0.786 | -0.46–2.04 | 0.369 |
|  | 2021 | 0.921 | -0.31–2.15 | 0.219 |
| 2019 | 2018 | -0.371 | -0.89–1.63 | 0.873 |
|  | 2020 | 1.157 | 0.01–2.31 | 0.048* |
|  | 2021 | 1.292 | 0.16–2.42 | 0.018* |
| 2020 | 2018 | -0.786 | -2.04–0.46 | 0.369 |
|  | 2019 | -1.157 | -2.31–-0.01 | 0.048* |
|  | 2021 | 0.135 | -0.99–1.26 | 0.990 |
| 2021 | 2018 | -0.921 | -2.15–0.31 | 0.219 |
|  | 2019 | -1.292 | -2.42–-0.16 | 0.018* |
|  | 2020 | -0.135 | -1.26–0.99 | 0.990 |

Diff = Difference; CI = Confidence Interval

*Statistically Significant

**Table 3. The 2018 and 2019 age-adjusted incidence rates per 100,000 population.**

| Age Group | Population | Case Count | IR per 100000 | 95% CI |
|---|---|---|---|---|
| **2018** | | | | |
| 15–19 | 147522 | 4 | 0.35 | 0.01–0.69 |
| 20–24 | 122281 | 24 | 2.45 | 1.47–3.43 |
| 25–29 | 105254 | 56 | 6.44 | 4.75–8.12 |
| 30–34 | 84615 | 112 | 15.35 | 12.51–18.20 |
| 35–39 | 76253 | 109 | 15.58 | 12.66–18.50 |
| 40–44 | 63984 | 83 | 12.97 | 10.18–15.76 |
| 45–49 | 53572 | 46 | 7.90 | 5.62–10.18 |
| 50–54 | 48136 | 21 | 3.58 | 2.05–5.11 |
| 55–59 | 31807 | 9 | 1.95 | 0.68–3.23 |
| 60–64 | 30074 | 1 | 0.19 | -0.18–0.56 |
| Total | 763498 | 465 | 66.77 | 60.63–72.90 |
| **2019** | | | | |
| 15–19 | 150169 | 6 | 0.51 | 0.10–0.93 |
| 20–24 | 124849 | 16 | 1.60 | 0.82–2.39 |
| 25–29 | 107465 | 68 | 7.65 | 5.84–9.48 |
| 30–34 | 86392 | 136 | 18.26 | 15.19–21.33 |
| 35–39 | 77854 | 144 | 20.16 | 16.87–23.45 |
| 40–44 | 65327 | 107 | 16.38 | 13.28–19.48 |
| 45–49 | 54697 | 55 | 9.25 | 6.81–11.69 |
| 50–54 | 49231 | 42 | 7.00 | 4.88–9.11 |
| 55–59 | 32475 | 4 | 0.85 | 0.02–1.68 |
| 60–64 | 30705 | 1 | 0.19 | -0.18–0.55 |
| Total | 779614 | 579 | 81.86 | 75.19–88.58 |

IR = Incidence Rate; CI = Confidence Interval

The annual incidence rates (per 100, 000) showed an increasing pattern as the years progressed. The line graph showed a steep slope from 2018 to 2019 and comparatively gentle slopes for the other years (**Fig 3**).

## Discussion

Age and early age at menarche are some of the contributing factors to the development of uterine fibroids as reported in literature [26]. Assessing the age of first diagnosis of uterine fibroids is essential as it helps to table and come out with interventions for the management of this disorder. In our study, the minimum age of first diagnosis was 17 years, and the maximum was 61 years, with an overall mean age of 36.29±8.08 years (**Table 1**). A study by Yu et al., in 2018, reported the mean age of the patients with new cases of uterine fibroids as 44.8 years with an age range of 18 to 65 years [17]. In another study by Myers et al. in the United States (US), the authors also reported a comparatively higher mean age of first diagnosis (40.9±5.9 years) [27]. This purports that uterine fibroids are diagnosed at a relatively younger age in our setting probably because of the increased awareness of uterine fibroids and the increased usage of ultrasound equipment in the country [9, 12]. In 2017, an in-service training program for ultrasound in midwifery was created by the Ghana Health Service (GHS), which has increased accessibility and early diagnosis of gynecological problems such as uterine fibroids and many others [28]. Ghana is also the headquarters of the Fibroid Foundation Africa, a multi-

**Table 4. The 2020 and 2021 age-adjusted incidence rates per 100,000 population.**

| Age Group | Population | Case Count | IR per 100000 | 95% CI |
|---|---|---|---|---|
| **2020** | | | | |
| 15–19 | 153783 | 8 | 0.67 | 0.21–1.14 |
| 20–24 | 127471 | 29 | 2.84 | 1.81–3.89 |
| 25–29 | 109721 | 98 | 10.81 | 8.67–12.95 |
| 30–34 | 88206 | 138 | 18.15 | 15.12–21.17 |
| 35–39 | 79489 | 169 | 23.17 | 19.68–26.66 |
| 40–44 | 66699 | 102 | 15.29 | 12.33–18.26 |
| 45–49 | 55846 | 47 | 7.74 | 5.62–9.96 |
| 50–54 | 50737 | 29 | 4.69 | 2.98–6.39 |
| 55–59 | 33157 | 9 | 1.87 | 0.65–3.10 |
| 60–64 | 31350 | 2 | 0.36 | -0.14–0.87 |
| Total | 796459 | 631 | 85.60 | 78.85–92.35 |
| **2021** | | | | |
| 15–19 | 163513 | 13 | 1.03 | 0.47–1.58 |
| 20–24 | 135964 | 52 | 4.78 | 3.48–6.08 |
| 25–29 | 122437 | 122 | 12.06 | 9.92–14.20 |
| 30–34 | 107692 | 177 | 19.07 | 16.26–21.87 |
| 35–39 | 96993 | 220 | 24.72 | 21.46–27.99 |
| 40–44 | 73181 | 89 | 12.16 | 9.64–14.69 |
| 45–49 | 62661 | 55 | 8.08 | 5.94–10.21 |
| 50–54 | 52347 | 47 | 7.36 | 5.26–9.47 |
| 55–59 | 41140 | 17 | 2.85 | 1.50–4.21 |
| 60–64 | 37775 | 2 | 0.30 | -0.11–0.72 |
| Total | 893703 | 794 | 92.40 | 85.88–98.92 |

IR = Incidence Rate; CI = Confidence Interval

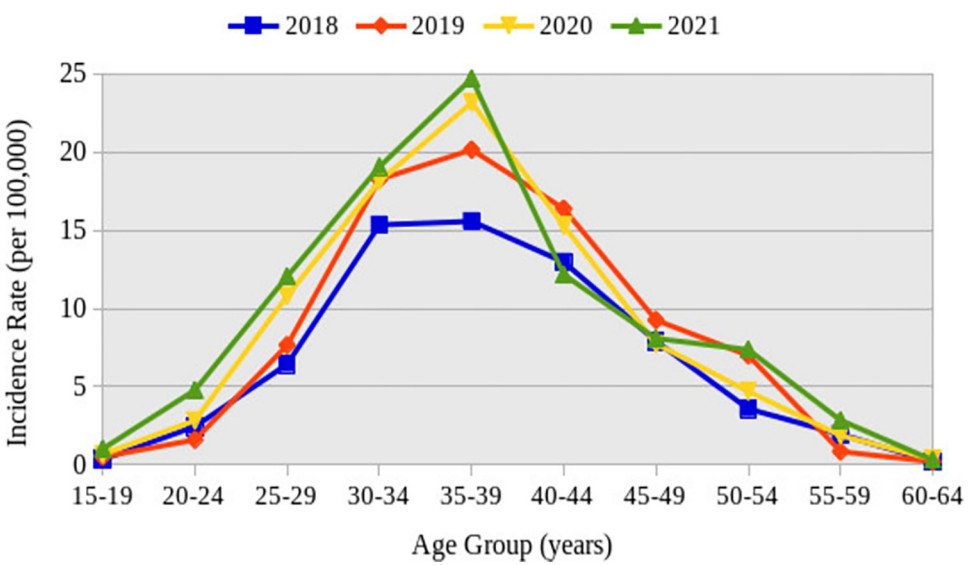

**Fig 2. Age-adjusted incidence rates of patients with first diagnosis of uterine fibroids.**

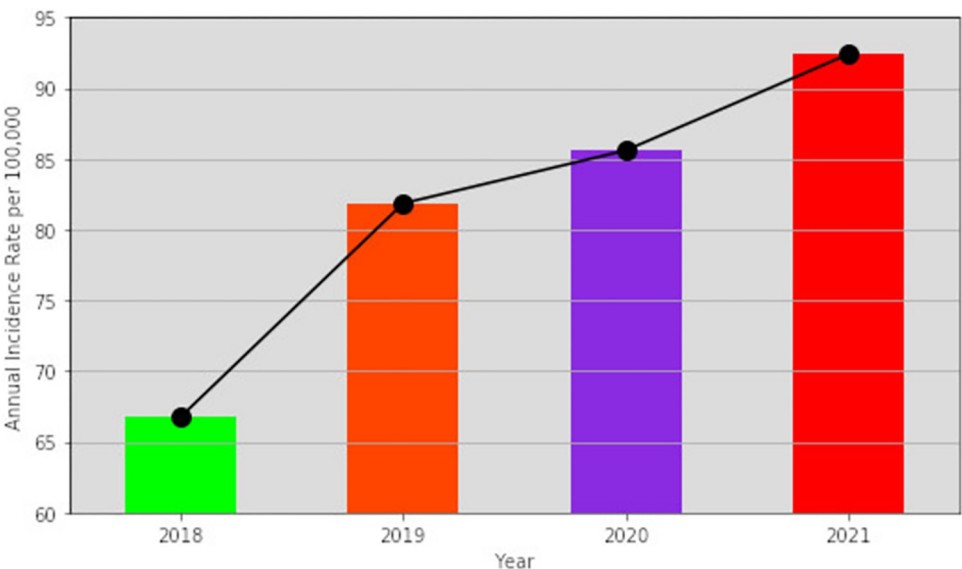

**Fig 3. Pattern of annual incidence rates of patients with first diagnosis of uterine fibroids.**

stakeholder charitable organization dedicated to creating awareness and encouraging more conversation about uterine fibroids thereby, increasing the awareness of this condition in the country and Africa at large [29]. A recent study has also reported routine check-ups as the leading indication for the diagnosis of uterine fibroids in Ghana [10]. The discrepancy accounting for the lower mean age in our setting as compared to the findings of the US could also be due to differences in population structures [23, 30]. This could also be as a result of the fact that the patients might have presented early with symptoms because of how common uterine fibroids are in Ghana from literature [21].

We observed a significant surge in the mean age of the patients with the first diagnosis of uterine fibroids in 2019 and significant decreases afterward in 2020 and 2021 (**Fig 1**). These could partly be due to an increase in awareness, routine screening, and advancement in technology, resulting in early diagnosis of uterine fibroids [29, 31, 32]. A study by Hira et al., highlighted the contribution of technological advancement in investigative tools when it comes to diagnosing tiny uterine fibroids, using high-frequency ultrasound probes that enhance diagnostic sensitivity, which are common in Ghana [9, 33, 34].

This current study found that, on average, about 617 new cases of uterine fibroids are recorded every year in Ghana (**Tables 3 and 4**). This figure is comparatively lower than what has been reported in another study by Yu et al., in 2018 [17]. In their study, the authors recorded an average of 1,381 new cases of uterine fibroids diagnosed each year in Washington. Stahlman et al. also reported an annual average incidence case of 1,224 among United States Army women [35]. The reason for the difference in the annual average numbers of newly diagnosed patients with uterine fibroids cannot be readily elucidated but may be as a result of exposure to endocrine disruptors during development as stated by a recent study in the United States [36]. A study by Prusinski et al., in the United States, brought to light that, exposure to the endocrine disruptor during development, increases the incidence of uterine fibroids by altering deoxyribonucleic acid (DNA) repair in myometrial stem cells, as the majority of the patients had this as a risk factor [36]. The authors added that the endocrine-disrupting chemicals, when exposed to, modify myometrial stem cells' ability to repair and reverse DNA damage, providing a driver for acquiring mutations that may promote the development of uterine

fibroids [36]. However, such genetic findings have not been reported in Ghana and were not evaluated in this study.

We reported an increasing pattern of annual incidence rates of uterine fibroids as the years went by, and a fluctuating trend across the age groups (**Figs 2 & 3**). The 35–39 years age group recorded the highest incidence rate (24.72 per 100,000) across all years in our study, followed by the 30–34 years age category (**Fig 2**). This may impact fertility as the most affected age groups belong to the prime reproductive age group (15–49 years) [37]. Stahlman et al. also recorded a high incidence rate in the 35–39 years age class (1774 per 100,000), which is similar to our finding [35]. On the contrary, Yu et al. found a fluctuating pattern and an overall decreasing trend whilst Stahlman et al., saw both steady and fluctuating patterns in the annual incidence rates [17, 35]. Yu et al. and Wise et al. both reported high incidence rates in the 45–49 years age class in almost all the years under study, followed by the 40–44 years age category, which is also contrary to what we found [17, 38].

Variable incidence rates have been reported in literature in various geographical settings. In this current study, we found that the annual incidence rate of uterine fibroids ranged from 66.99–92.40 per 100,000 women (95%CI = 60.63–98.92). Stewart et al. also found incidence rates ranging from 217–3745 cases per 100,000 women in Europe and America, which is higher [4]. In Germany, a lower annual incidence rate of 12.7 per 100,000 women was reported by Soliman et al. [39]. In other literature, the incidence rate was reported to be 3440 per 100,000 (95%CI = 3310–3570) women in the US and 580 per 100,000 women in the United Kingdom [35, 40, 41]. The report of a higher incidence of uterine fibroids in many advanced countries compared to our setting may be a result of the availability of more human and infrastructural resources coupled with state-of-the-art diagnostic equipment.

The significant annual increase in the number of new cases of uterine fibroids among women of all ages and races across the world has been reported by a number of studies in time past, as also seen in our study [13, 42]. In Africa, especially those of Negroid race, Geidan et al. reported an upsurge in the number of hysterectomies resulting from the high incidence of uterine fibroids [42]. This poses lots of threats to the quality of life of women, hence the need to set out interventions to reduce the associated risk factors, which turn out to also influence the available treatment measures and severity of fibroid outcome [13, 43, 44].

## Limitation and strength of the study

Patients whose three years past medical records prior to the cohort years could not be obtained, were not included in the study and this could potentially reduce the case count, which is a notable limitation. However, including all cases (who met the inclusion criteria) will statistically be representative of the population. To the best of our knowledge and from review of literature, trends in uterine fibroids diagnoses by age/years were not previously available in our setting. Again, assessing the age of first diagnosis of uterine fibroids is essential as it helps to table and come out with interventions for the management of this disorder.

## Conclusion

The incidence rate of uterine fibroids showed an annual increasing trend and it was highest in the 35–39 years age category over study period. The most affected age group is the prime reproductive group, therefore the occurrence of uterine fibroids may impact on fertility. Despite the significant decreasing trend in the annual mean ages of first diagnosis of uterine fibroids suggestive of early diagnoses, it is germane for health practitioners, and health policy makers to put in place measures to cater for the increases in order to prevent complications associated with uterine fibroids.

## Supporting information

**S1 Dataset.**
(XLSX)

## Acknowledgments

The management of CCTH and the staff of the records department of the CCTH are appreciated for their support in making this study successful.

## Author Contributions

**Conceptualization:** Emmanuel Kobina Mesi Edzie, Klenam Dzefi-Tettey, Edmund Kwakye Brakohiapa, Sebastian Ken-Amoah.

**Data curation:** Emmanuel Kobina Mesi Edzie, Klenam Dzefi-Tettey, Edmund Kwakye Brakohiapa, Sebastian Ken-Amoah.

**Formal analysis:** Frank Quarshie, Obed Cudjoe, Evans Boadi, Joshua Mensah Kpobi, Richard Ato Edzie.

**Investigation:** Henry Kusodzi, Prosper Dziwornu, Abdul Raman Asemah.

**Methodology:** Emmanuel Kobina Mesi Edzie, Klenam Dzefi-Tettey, Edmund Kwakye Brakohiapa, Frank Quarshie, Sebastian Ken-Amoah, Obed Cudjoe, Evans Boadi, Joshua Mensah Kpobi, Richard Ato Edzie.

**Supervision:** Emmanuel Kobina Mesi Edzie.

**Validation:** Henry Kusodzi, Prosper Dziwornu, Abdul Raman Asemah.

**Visualization:** Frank Quarshie, Obed Cudjoe, Evans Boadi, Joshua Mensah Kpobi, Richard Ato Edzie.

**Writing – original draft:** Emmanuel Kobina Mesi Edzie, Klenam Dzefi-Tettey, Edmund Kwakye Brakohiapa, Frank Quarshie, Sebastian Ken-Amoah, Obed Cudjoe, Evans Boadi, Joshua Mensah Kpobi, Richard Ato Edzie, Henry Kusodzi, Prosper Dziwornu, Abdul Raman Asemah.

**Writing – review & editing:** Emmanuel Kobina Mesi Edzie, Klenam Dzefi-Tettey, Edmund Kwakye Brakohiapa, Frank Quarshie, Sebastian Ken-Amoah, Obed Cudjoe, Evans Boadi, Joshua Mensah Kpobi, Richard Ato Edzie, Henry Kusodzi, Prosper Dziwornu, Abdul Raman Asemah.

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
