## [Decision Letter · Decision Letter 0]

15 Dec 2022

PONE-D-22-20619Age of First Diagnosis and Incidence Rate of Uterine Fibroids in Ghana. A Retrospective Cohort Study.PLOS ONE

Dear Dr. EDZIE,

Thank you for submitting your manuscript to PLOS ONE. After careful consideration, we feel that it has merit but does not fully meet PLOS ONE’s publication criteria as it currently stands. Therefore, we invite you to submit a revised version of the manuscript that addresses the points raised during the review process.

ACADEMIC EDITOR:  The manuscript is a beneficial study examining the age-adjusted incidence and preferred age of onset of uterine fibroids in Ghana, Africa, but it lacks novelty. There is some text suggesting the possibility of early diagnosis as to why the incidence is increasing, but the background, for example, the increased use of ultrasound equipment in Ghana or the introduction of MRI scans, is unclear, so it is ultimately difficult to find a basis for the claims made. Also, although the initial diagnosis is discussed, it is difficult to understand whether this is because the public's understanding of uterine fibroids has improved or because check-ups have become more frequent, as there is no information on gynecological check-ups in the country. The background and discussion sections require significant revision. We look forward to receiving your revised vesion in soon.

We look forward to receiving your revised manuscript.

Kind regards,

Kazunori Nagasaka

Academic Editor

PLOS ONE

Journal Requirements:

2. Please ensure that you have specified (1) whether consent was informed and (2) what type you obtained (for instance, written or verbal, and if verbal, how it was documented and witnessed). If your study included minors, state whether you obtained consent from parents or guardians. If the need for consent was waived by the ethics committee, please include this information.

3. You indicated that you had ethical approval for your study. In your Methods section, please ensure you have also stated whether you obtained consent from parents or guardians of the minors included in the study or whether the research ethics committee or IRB specifically waived the need for their consent.

Additional Editor Comments:

This is a very useful study examining age-adjusted incidence and preferred age of onset of uterine fibroids in Ghana, Africa, but lacks novelty. There is some text suggesting the possibility of early diagnosis as to why the incidence is increasing, but the background, for example, the increased use of ultrasound equipment in Ghana or the introduction of MRI scans, is unclear, so it is ultimately difficult to find a basis for the claims made. Also, although the initial diagnosis is discussed, it is difficult to understand whether this is because the public's understanding of uterine fibroids has improved or because check-ups have become more frequent, as there is no information on gynaecological check-ups in the country. The background and discussion sections require significant revision.

Reviewers' comments:

Reviewer's Responses to Questions

**Comments to the Author**

1. Is the manuscript technically sound, and do the data support the conclusions?

Reviewer #1: Yes

2. Has the statistical analysis been performed appropriately and rigorously? 

Reviewer #1: Yes

3. Have the authors made all data underlying the findings in their manuscript fully available?

Reviewer #1: Yes

4. Is the manuscript presented in an intelligible fashion and written in standard English?

Reviewer #1: No

5. Review Comments to the Author

Reviewer #1: Age of First Diagnosis and Incidence Rate of Uterine Fibroids in Ghana. A Retrospective Cohort Study.

Title

• Please review the title, the study design and what was done do not match. Is this relay a cohort study?

English language check

• Language is not clear some sentences could be paraphrased. Please use clear unambiguous professional English throughout the manuscript.

Abstract

• The first sentence should be paraphrased (line 37-39).

• Please stick to using one term throughout the text for consistency and easy flow of ideas (uterine fibroids). Other terms can be introduced and left as such.

• Clearly describe this as a disorder

• Our setting (Line 40)- be specific is it rural or urban?

Introduction

• The introduction is not well written.

• Provide some few statistics on what is happening globally, in Africa and in Ghana to give a clear picture of what is currently happening.

• Line 66-67-the sentences is not clear, paraphrase.

• Line 67-68 work on the flow of ideas, this does not read well.

• Line 76-77 be more specific, is it possible to link the sentence to line 64-65.

• Paraphrase line 82-84.

• The aim could be highlighted or overemphasized for clarity.

Methods

• Describe the study design fully

• Paraphrase line 106-108, “assessed” is not the correct term to use.

• Line 116-118 restructure the sentence does not read well.

• Line 123-126 paraphrase

• Line 127 “consecutively selected” use a better term

• Justify why this is a cohort study

Ethics

• Line 156 please give the official name of the ethical review committee.

Results

• For tables that do not fit in one page enable repeat table titles.

• Line 164-165, line 181“the rests” paraphrase

• Results are fairly well presented.

Discussion

• Line 213-214 add references

• Line 276-277 not clear

• Reference lines 234,239 and 241.

• Consider restructuring the discussion and paraphrasing the sentences for easy flow of ideas.

• Note that there is comparability of the studies referenced, your study was only covering South Central Ghana and most of the studies were national.

Limitations

• Not fully explained.

• What are the strengths of the study?

Conclusion

• It is fairly well stated and linked to the aim.

General comment

• The writing style and the English language can be improved to ensure that the paper can be clearly understood by the international audience, the current phrasing makes comprehension difficult.

• Citations can be presented better to help with the flow of ideas and to avoid repetition.

Reference

• Consistency in presentation (i.e. highlighting and underlining)

6. PLOS authors have the option to publish the peer review history of their article (what does this mean?). If published, this will include your full peer review and any attached files.

Reviewer #1: No

---

## [Author Response · Author response to Decision Letter 0]

27 Dec 2022

Editor-in-Chief

PLOS ONE 

 27/12/2022

Dear Editor,

RESPONSES TO THE COMMENTS OF THE REVIEWERS FOR THE MANUSCRIPT “Age of First Diagnosis and Incidence Rate of Uterine Fibroids in Ghana. A Retrospective Cohort Study.” WITH MANUSCRIPT NUMBER: PONE-D-22-20619.

We are grateful to the reviewers for their valuable efforts in reviewing this manuscript and we believe their in-depth reviews and comments will help improve on the quality and the structure of this manuscript for possible publication in this prestigious Journal. We also thank the Editor for the constructive comments and for his/her time and efforts in handling this manuscript.

Please find below the comments from the reviewers and the corresponding responses based on the numbers assigned to them.

EDITOR

Comment: There is some text suggesting the possibility of early diagnosis as to why the incidence is increasing, but the background, for example, the increased use of ultrasound equipment in Ghana or the introduction of MRI scans, is unclear, so it is ultimately difficult to find a basis for the claims made.

Response: The background and the discussion sections of the study has been revised to reflect the basis for the claims made. We are very grateful for this constructive critique.

Comment: Also, although the initial diagnosis is discussed, it is difficult to understand whether this is because the public's understanding of uterine fibroids has improved or because check-ups have become more frequent, as there is no information on gynecological check-ups in the country.

Response: The background and the discussion sections of the study has been revised to reflect the basis for the claims made. We are very grateful for this constructive critique.

Comment: You indicated that you had ethical approval for your study. In your Methods section, please ensure you have also stated whether you obtained consent from parents or guardians of the minors included in the study or whether the research ethics committee or IRB specifically waived the need for their consent.

Response: The Methods section has been revised to include the informed consent obtained from the parents of the minor in the study. We are thankful for this comment.

REVIEWER

Abstract

Comment: Please review the title, the study design and what was done do not match. Is this relay a cohort study?

Response: This is a cohort study. Cohort studies are the best method for determining the incidence as done in this current study, and natural history of a condition. The studies may be prospective or retrospective and sometimes two cohorts are compared. Cohort analysis treats an outcome variable as a function of cohort membership, age, and period. For a population at any given moment, there are three explanatory temporal dimensions: age, period, and cohort effects. From our study, we carried out a population-based retrospective cohort study with complete data on female enrollees for the most recent 4-year period (Jan. 1, 2018, through Dec. 31, 2021). For 2018, the cohort comprised 763498 women of observation and 465 new fibroid cases were observed. For 2019, the cohort consisted 779614 women of observation and 579 new fibroid cases were observed. For 2020, the cohort comprised 796459 women of observation and 631 new fibroid cases were observed and For 2021, the cohort comprised 893703 women of observation and 794 new fibroid cases were observed. Detailed report on this as well the age-adjusted incidence rates are in Tables 3 & 4. Below are some references for the assertions above;

1. Mann CJ. Observational research methods. Research design II: cohort, cross sectional, and case-control studies. Emergency medicine journal. 2003 Jan 1;20(1):54-60.

2. W.M. Mason, N.H. Wolfinger, in International Encyclopedia of the Social & Behavioral Sciences, 2001

3. S.D. Withers, in International Encyclopedia of Human Geography (Second Edition), 2009

Comment: Language is not clear some sentences could be paraphrased. Please use clear unambiguous professional English throughout the manuscript.

Response: Some sentences have been paraphrased to make comprehension easier. Thank you for this constructive comment.

Comments: The first sentence should be paraphrased (line 37-39).

Response: Line 37-39 has been paraphrased. Thank you very much.

Comment: Please stick to using one term throughout the text for consistency and easy flow of ideas (uterine fibroids). Other terms can be introduced and left as such.

Response: We thank the reviewer for this comment. Changes have been made to reflect this new development.

Comments: Clearly describe this as a disorder

Response: We are thankful for this comment. A description of this disorder has been added.

Comments: Our setting (Line 40)- be specific is it rural or urban?

Response: The specific setting has been added. Thank you very much.

Introduction

Comment: Provide some few statistics on what is happening globally, in Africa and in Ghana to give a clear picture of what is currently happening.

Response: We are grateful for this constructive comment. Some information has been provided globally (Line 64-65).

Lines 72-77 also give a summarize statistical information about uterine fibroids in America (North and South), Europe, Asia and Africa, and referenced accordingly.

Lines 77-79 also indicated some statistics on uterine fibroids in Sub-Saharan Africa, also referenced.

Lines 80-85 give a clear picture of what is currently happening in Ghana, also appropriately referenced.

Comment: Line 66-67-the sentences is not clear, paraphrase.

Response: The paraphrasing has been done. Thank you very much.

Comment: Line 67-68 work on the flow of ideas, this does not read well.

Response: The paraphrasing has been done. Thank you very much.

Comment: Line 76-77 be more specific, is it possible to link the sentence to line 64-65.

Response: We are grateful for this comment. More specific statistics have been added to line 76-77 as suggested by the reviewer.

Comment: Paraphrase line 82-84.

Response: The paraphrasing has been done. Thank you very much.

Comment: The aim could be highlighted or overemphasized for clarity.

Response: The specific aim has been highlighted for clarity. Thank you very much.

Methods

Comment: Describe the study design fully

Response: The study design has been elaborated as per the reviewer’s suggestion. Thank you very much.

Comment: Paraphrase line 106-108, “assessed” is not the correct term to use.

Response: The paraphrasing has been done and a correct term has been used. Thank you very much.

Comment: Line 116-118 restructure the sentence does not read well.

Response: Line 116-118 has been restructured to read well. Thank you very much

Comment: Line 123-126 paraphrase

Response: The paraphrasing has been done. Thank you very much.

Comment: Line 127 “consecutively selected” use a better term

Response: Purposive has been used since cases were selected because they have characteristics that we need for the study.

Ethics

Comment: Line 156 please give the official name of the ethical review committee.

Response: The official name for the ethical review committee has been added to the revised manuscript. We appreciate this comment.

Results

Comment: For tables that do not fit in one page enable repeat table titles.

Response: We are grateful for this constructive comment. Tables have been made to fit one page.

Comment: Line 164-165, line 181 “the rests” paraphrase

Response: This portion has been paraphrased accordingly. Thank you very much.

Comment: Results are fairly well presented.

Response: We are grateful for the good appraisal of the presentation of results portion.

Discussion

Comment: Line 213-214 add references

Response: Reference for line 213-214 has been add accordingly. Thanks for the comment.

Comment: Line 276-277 not clear

Response: Some paraphrasing has been done. Thank you very much.

Comment: Reference lines 234,239 and 241

Response: References for lines 234,239 and 241 have been add accordingly. Thanks for the comment.

Comment: Consider restructuring the discussion and paraphrasing the sentences for easy flow of ideas.

Response: Required restructuring of the discussion and paraphrasing sentences for easy flow of ideas have been done. We are grateful for this comment.

Comment: Note that there is comparability of the studies referenced, your study was only covering South Central Ghana and most of the studies were national.

Response: We are grateful for this comment. However, per our search, most of the studies compared to our finding were carried out in specific states/regions within a country. For instance, Yu et al.’s study in the USA was done in Washington. Stahlman et al., Myers et al., and Wise et al.’s study were also not national, and many others. 

Again, we made use of the direct method of standardization founded on the World Health Organization (WHO) population distribution standard, and this allows for comparison across populations with different age composition. Referenced below;

World Health Organization (WHO). Age Standardization of Rates: A New WHO Standard. 2001. Available from: https://www.who.int/healthinfo/paper31.pdf

Limitations

Comment: Not fully explained.

Response: This portion has been paraphrased to make the meaning clearer.

Comment: What are the strengths of the study?

Response: The strengths of the study have added to the revised manuscript. We appreciate this constructive comment.

Conclusions

Comment: It is fairly well stated and linked to the aim.

Response: We are grateful for the good appraisal of the conclusion section.

General Comments

Comment: The writing style and the English language can be improved to ensure that the paper can be clearly understood by the international audience, the current phrasing makes comprehension difficult.

Response: The writing style and the English language have been improved to ensure that the paper can be clearly understood by the international audience. Thank you for this constructive appraisal.

Comment: Citations can be presented better to help with the flow of ideas and to avoid repetition.

Response: We are grateful for this comment. The citations have been presented better to improve the flow of ideas.

References

Comment: Consistency in presentation (i.e. highlighting and underlining)

Response: Consistency has been ensured. We are grateful for this comment.

All of the responses and changes have been incorporated into the revised manuscript and highlighted in yellow.

Kind Regards.

Dr. Emmanuel Kobina Mesi Edzie.

Corresponding Author

---

## [Decision Letter · Decision Letter 1]

6 Mar 2023

Age of First Diagnosis and Incidence Rate of Uterine Fibroids in Ghana. A Retrospective Cohort Study.

PONE-D-22-20619R1

Dear Dr. EDZIE,

We’re pleased to inform you that your manuscript has been judged scientifically suitable for publication and will be formally accepted for publication once it meets all outstanding technical requirements.

Kind regards,

Kazunori Nagasaka

Academic Editor

PLOS ONE

Additional Editor Comments (optional):

Dear Authors,

Thank you very much for sending us your responses and the revised manuscript.

I think the manuscript is well-improved and acceptable for publication on Plos One.

Congratulations, and we look forward to your future manuscript.

Reviewers' comments:

Reviewer's Responses to Questions

**Comments to the Author**

1. If the authors have adequately addressed your comments raised in a previous round of review and you feel that this manuscript is now acceptable for publication, you may indicate that here to bypass the “Comments to the Author” section, enter your conflict of interest statement in the “Confidential to Editor” section, and submit your "Accept" recommendation.

Reviewer #1: All comments have been addressed

2. Is the manuscript technically sound, and do the data support the conclusions?

Reviewer #1: Yes

3. Has the statistical analysis been performed appropriately and rigorously? 

Reviewer #1: Yes

4. Have the authors made all data underlying the findings in their manuscript fully available?

Reviewer #1: No

5. Is the manuscript presented in an intelligible fashion and written in standard English?

Reviewer #1: Yes

6. Review Comments to the Author

Reviewer #1: Age of First Diagnosis and Incidence Rate of Uterine Fibroids in Ghana. A Retrospective Cohort Study.

Abstract

• Line 39. The incidence rate is high among women of all races and ages, needing much attention.

-Consider paraphrasing the above sentence.

Introduction

• The introduction is well written.

• The objectives are very clear.

Results

Table 1. Socio-demographics of the Participants and Annual Mean Age Comparison.

- Please consider presenting age in one row to save space.

- e.g Age 17-61 (36.29 (8.08) years)

7. PLOS authors have the option to publish the peer review history of their article (what does this mean?). If published, this will include your full peer review and any attached files.

Reviewer #1: No

---

## [Editor Report · Acceptance letter]

9 Mar 2023

PONE-D-22-20619R1 

Age of First Diagnosis and Incidence Rate of Uterine Fibroids in Ghana. A Retrospective Cohort Study. 

Dear Dr. Edzie:

I'm pleased to inform you that your manuscript has been deemed suitable for publication in PLOS ONE. Congratulations! Your manuscript is now with our production department. 

Kind regards, 

on behalf of

Professor Kazunori Nagasaka 

Academic Editor

PLOS ONE